# In-Feed Supplementation of Resin Acid-Enriched Composition Modulates Gut Microbiota, Improves Growth Performance, and Reduces Post-Weaning Diarrhea and Gut Inflammation in Piglets

**DOI:** 10.3390/ani11092511

**Published:** 2021-08-26

**Authors:** Md Karim Uddin, Shah Hasan, Md. Rayhan Mahmud, Olli Peltoniemi, Claudio Oliviero

**Affiliations:** 1Department of Production Animal Medicine, Faculty of Veterinary Medicine, University of Helsinki, 04920 Saarentaus, Finland; shah.hasan@helsinki.fi (S.H.); rayhan.mahmud@helsinki.fi (M.R.M.); olli.peltoniemi@helsinki.fi (O.P.); claudio.oliviero@helsinki.fi (C.O.); 2Hankkija Oy, Peltokuumolantie 4, 05801 Hyvinkää, Finland

**Keywords:** resin acid, piglet, gut microbiota, post weaning diarrhea, growth performance, myeloperoxidase

## Abstract

**Simple Summary:**

The weaning process can be detrimental to piglets and often leads to low feed intake, low weight gain, diarrhea, and eventually increased mortality. Resin acid-enriched composition (RAC) is an innovative natural feed ingredient that contains 90% fatty acids, mostly linoleic and oleic, pinolenic acids, and conjugated linoleic acid (CLA). It also contains 9% active component resin acid (RA), mostly abietic and dehydroabietic, and pimaric acids. It possesses antimicrobial, anti-inflammatory, and immunomodulatory properties and has been used in feed to improve performance in pigs. We hypothesized that RAC supplementation to sow and piglet diets might modulate intestinal microbiota, and improve post-weaning growth and reduce post-weaning diarrhea (PWD) and gut inflammation. Sow diets were supplemented with RAC from the last six weeks of gestation until weaning, and piglets were fed RAC trough creep feed and post weaning feed until seven weeks of age. Results indicated that supplementing sow and piglet diets with RAC during late gestation, lactation and at post weaning modulates gut microbiota, reduces inflammatory biomarkers, improves growth performance of post-weaning piglets, and prevents PWD in weaned piglets. Considering these results, RAC supplementation represents a potential strategy to improve piglet growth performance, and is a natural alternative to antibiotics for preventing PWD.

**Abstract:**

The weaning process represents a delicate phase for piglets, and is often characterized by lower feed intake, lower weight gain, diarrhea, and ultimately increased mortality. We aimed to determine the effects of RAC supplementation in diets on improving piglet growth and vitality, reducing post-weaning diarrhea, and enhancing gut health. In a 2 × 2 × 2 factorial experiment, we selected forty sows and their piglets. Piglets were followed until seven weeks of age. There were no significant differences found between RAC treated and control piglets until weaning (*p* = 0.26). However, three weeks after weaning, RAC treated piglets had higher body weight and average daily growth (ADG) than the control piglets (*p* = 0.003). In addition, the piglets that received RAC after weaning, irrespective of mother or prior creep feed treatment, had lower post-weaning diarrhea (PWD) and fecal myeloperoxidase (MPO) level than control piglets. Gut microbiota analysis in post-weaning piglets revealed that RAC supplementation significantly increased Lachnospiraceae_unclassified, *Blautia*, *Butyricicoccus*, *Gemmiger* and *Holdemanella*, and decreased Bacteroidales_unclassified. Overall, RAC supplementation to piglets modulated post-weaning gut microbiota, improved growth performance after weaning, reduced post-weaning diarrhea and reduced fecal myeloperoxidase levels. We therefore consider RAC to be a potential natural feed supplement to prevent enteric infections and improve growth performance in weaning piglets.

## 1. Introduction

Weaning is a critical stage in piglets’ lives since they are predisposed to various gastrointestinal illnesses and impaired performance [1]. During the weaning process, piglets are at risk of low feed intake, minimal weight gain, diarrhea, and increased mortality [2,3]. This challenging process is attributable to the immature gut structure of the piglets, which are unable to digest solid food particles fully, and the fermented undigested food promotes the growth of pathogenic bacteria in the gut [4]. Hence, piglets exhibit malabsorption syndrome followed by increased excretion of fatty acids and watery feces, degenerated intestinal villi, inflammation [5], and impaired immune functions [6]. In most cases, opportunistic bacteria, such as enterotoxigenic *Escherichia coli* (ETEC) aggravate this condition and can cause post-weaning diarrhea (PWD). PWD has serious effects on weaned piglets, leading to increased mortality and production losses [7]. In Europe, PWD incidence increased after the ban on antibiotics used as growth promoters in animal feed in 2006 [8]. Preventive use of high-level zinc oxide in weaned piglets’ diets has been used as an alternative to control ETEC outbreaks in piglets [9]. However, the use of this compound will also be banned by EU legislation starting from June 2022 [10,11]. With the increasing incidence of PWD and resistance to antibiotics by enterotoxigenic bacterial strains, developing alternative feeding strategies to prevent infections on farms is important [12]. Sow-feeding strategies include, at different stages of production, the use of fibers, prebiotics, probiotics, essential oils, fatty acids, organic acids, and resin acid compounds [13,14,15]. Resin acid-enriched composition (RAC) is an innovative feed additive derived from coniferous trees and produced by thermal distillation without added chemicals from a side stream of wood pulp manufacture [15,16]. It contains 90% fatty acids (mostly linoleic, oleic, pinolenic acids, conjugated linoleic acid (CLA), and 9% active component resin acid (RA) (mostly abietic, dehydroabietic, and pimaric acids) [15].

Some in vitro studies reported RA to exert substantial antimicrobial activity against gram-positive bacteria [17,18], mild activity against gram-negative bacteria [19], and being highly selective against bacterial membranes rather than mammalian cells [20]. In addition, RAs, mostly abietic acid and dehydroabietic acid derivatives, have antibacterial, antifungal, and antiparasitic properties [21,22]. Studies in humans and animals established that RAC components such as abietic acids, dehydroabietic acids, CLA, and pinolenic acids all have anti-inflammatory and immunomodulatory effects [23,24]. Recent studies suggested that RAC in sow feed can improve sow performance at farrowing and early lactation, increasing the colostrum IgG content, and modulating the gut microbiota [15].

Insufficient colostrum intake has been identified as one of the major causes of neonatal mortality in modern pig production [25], especially for low-birth-weight piglets [26,27]. Low-birth-weight piglets are less vital and find it difficult to compete with littermates to gain access to the udder for sufficient colostrum intake [27,28]. Piglet vitality refers to the strength and vigor of piglets, which is necessary for their survival [29]. Several studies have used different piglet traits to assess piglet vitality [29,30]. Moreover, several studies investigated the effects of dietary supplementation in late gestating sows on improving piglet vitality and colostrum intake [31], thus revealing the beneficial effects of late gestation supplementation of active feed ingredients on sows and piglet vitality.

Gut microbiota are regarded as one of the key regulators of piglet health, especially for weaning piglets [32]. Before weaning, maternal and environmental microbiota shape the gut microbiota of piglets [33]. However, the ingestion of solid feeds during the suckling-to-weaning transition period induces a “weaning-reaction”, caused by a transient intestinal immune response, which renders the piglets susceptible to gut inflammation and metabolic dysfunction [34,35], resulting in reduced production performance [7]. There have been different studies conducted, mainly focusing on describing the taxonomy and functional capacity of gut microbiota after weaning [36,37], but there are only few studies on the effects of dietary treatment in reducing gut inflammation and improving production performance of post-weaning piglets by modulating beneficial metabolites that produce the microbiota. Our previous research showed that RAC supplementation in sows can modulate gut microbiota [15]. Other research showed that RAC supplementation improves the microbial population and growth performance of broilers [38,39].

Additionally, we aimed to investigate the indirect maternal effects (when RAC is provided to sows during late pregnancy) and the direct dietary effects of RAC on piglets (when RAC is fed to piglets in the creep feed and post weaning). Considering RAC’s antibacterial, anti-inflammatory, and immunomodulatory effects, we hypothesized that RAC might modulate the intestinal microbiota, improve post-weaning growth, and reduce PWD and gut inflammation. Moreover, we aimed to determine the effects of RAC supplementation on sows and piglets regarding piglet vitality, gut microbiota, growth performance, post-weaning diarrhea incidence, and fecal inflammatory biomarkers MPO.

## 2. Materials and Methods

The entire experiment was carried out on a commercial pig farm in Kauhava, western Finland, from April 2020 to September 2020 (ELLA; ESAVI/2325/04.10.07/2017 with modification ESAVI/17315/2020). In our experiment sows farrowed in different batches during this period.

### 2.1. Animals and Experimental Design

#### 2.1.1. Sow Selection and Feeding

In a standard loose housed pregnancy room, 40 multiparous sows, belonging to the Figen Muscle synthetic sire line (Figen Oy, Pietarsaari, Finland) developed from Finnish Landrace and Finnish Large White breeds, balanced for body condition (back fat thickness) and parity (2–4), were allocated a standard diet (Appendix A) differing only for RAC 0.15% (Progres^®^ from Hankkija Oy, Hyvinkää, Finland, Appendix A) supplementation for the treatment sows (*n* = 20) started six weeks before expected farrowing date. After moving to the farrowing room seven days before expected farrowing, both control and treatment sows were fed a standard lactation diet (Appendix A), and only the 20 RAC sows continued with the RAC (0.15% RAC in lactation diet) supplementation until weaning (Figure 1).

#### 2.1.2. Pre-Weaning/Early Piglet Feeding

At farrowing every piglet in each litter was ear tagged based on birth order. During lactation, half of the litters (RR; *n* = 10) from RAC treated sows, received creep feed (Appendix A) containing RAC (0.1% RAC), while the remaining litters (RC; *n* = 10) got only the control creep feed. Similarly, half of the litters (CR; *n* = 10) from CONTROL sows received RAC treated creep feed, and the other half (CC; *n* = 10) got only the control creep feed (Figure 1).

#### 2.1.3. Post-Weaning Piglets Feeding

At weaning (28 days), 285 piglets from four lactation groups (RR = 67, RC = 82, CR = 67, and CC = 69) were included in a post-weaning piglet feeding trial. All of the different lactating groups were divided into four post-weaning RAC treatment (0.1% RAC in post-weaning feed, Appendix A) and four control diet, generating eight different treatments (RRR = 32, RRC = 35, RCR = 33, RCC = 49, CRR = 31, CRC = 36, CCR = 39, CCC = 30), which were followed until seven weeks of age (Figure 1).

### 2.2. Parameters and Measurements

Immediately after birth, the vitality (V) of piglets was visually assessed through breathing and locomotor behaviors and scored using a categorical 0–3 scale as described by Baxter et al. [31]. When there is no movement and no breathing after 15 s, vitality score was 0; when there was no movement after 15 s but the piglet was breathing or attempting to breathe, vitality score was 1; when the piglet showed little movement within 15 s, breathing or attempting to breathe, vitality score was 2; when the piglet had good movement, good breathing, and it attempted to stand within 15 s, vitality score was 3. Based on vitality scores, a score of zero was for dead piglets. We subsequently categorized our piglets as vitality 1 and vitality 2 since there were no piglets with a score of 3. All the piglets were individually weighed at birth, 24 h after birth, at weaning (four weeks of age), and post-weaning (seven weeks of age). The colostrum yield (CY) was calculated by the summation of the colostrum intake (CI) of individual piglets within a litter by using the following regression formula of Theil et al. [40].
CI, g = −106 + 2.26 WG + 200 BWB + 0.111 D − 1414 WG/D + 0.0182(1)

WG/BWB, where weight gain (WG) is stated in grams, body weight at birth (BWB) is stated in kg, and D is the duration of colostrum suckling (time after birth to until 24 h after the first piglet was born) stated in minutes. Observed sow parameters were parity, gestation length, farrowing duration (time of first piglet birth to last piglet birth) and numbers of total born, live born and stillborn piglets. Recorded piglet parameters were birth interval, pre-weaning mortality, post-weaning mortality, BWB, body weight at 24 h (BW24), weaning body weight, body weight at post-weaning, average daily gain (ADG), and diarrhea prevalence at weaning (28 days) and post-weaning (49 days). Piglets were divided into three categories based on their body weight at birth (BWB), such as low BW piglets (<1.0 kg), medium BW piglets (1.0–5 kg) and high BW piglets (>1.5 kg). A veterinarian recorded all the diarrhea cases visually during fecal sample collection at weaning and post-weaning. For practical reasons, due to cross-fostering, some piglets (*n* = 117) were excluded from the study between day 1 and day 28 of lactation in order to keep litters intact and attached to their original mother.

#### 2.2.1. Sample Collection

##### Piglet Sample Collection

Immediately after birth, we marked piglets on their back according to their birth order, and body weight was measured using a digital scale (XL-Float-22, Patriot^®^, Hyvinkää, Finland). The piglet’s back was gently soaked with a paper towel before marking and it was put back in the same place where it was taken from. All piglets were weighed again after 24 h of the first piglet’s birth to calculate colostrum intake as described by Theil et al. [40], and then ear tagged. All the farrowing activities were monitored and recorded using Denver IP cameras (SHO-110, Denver Electronics^®^, Hinnerup, Denmark). Piglets were followed until seven weeks of age, three weeks after weaning. Fecal samples were collected from six piglets of every litter at four weeks (*n* = 120 RAC; *n* = 117 CON), and at seven weeks of age those were included in the post-weaning trial (*n* = 80 RAC; *n* = 80 CON) using sterile swabs and 5 mL Eppendorf tubes. During collection samples were kept in an icebox and then stored at −20 °C.

##### Sow Sample Collection

We collected 20 mL of colostrum from the first three teats of same side of the anterior udder of each sow within the first 2 h after the first piglet birth [15]. After collection of colostrum samples, 0.3 mL colostrum was used to measure the colostrum brix value by a method described by Hasan et al. [41], using a commercial digital brix refractometer (digital hand-held pocket refractometer; Atago, Tokyo, Japan), with a range of 0 to 53% brix. Then samples were divided into three aliquots and stored at −20 °C until further analysis.

### 2.3. Laboratory Analysis

#### 2.3.1. Pig Myeloperoxidase ELISA

Fecal myeloperoxidase concentrations were measured using a commercial ELISA-kit (Pig Myeloperoxidase, CSB E 09397p, Cusabio, Wuhan, China) according to the manufacturer’s instructions. Fecal samples (*n* = 400) were prepared for analysis by placing 0.1 to 0.3 g into a 2 mL Eppendorf tube and diluting the sample by adding buffer (50 mM Tris-HCl, pH 8.0) so that the dilution was 1:5. Samples were homogenized for 5 min with the vortex mixer. If samples had not disintegrated, they were kept in a fridge for 1 h and manually mixed using a plastic rod. After homogenization samples were centrifuged at 18,000 g for 10 min at 4 °C. The supernatant was further diluted 1:10 with the sample diluent buffer provided with the ELISA kit.

#### 2.3.2. Gut Microbiota Sequencing

We extracted microbial DNA from 250 mg feces of each sample using the DNeasy PowerSoil Pro Kit (Qiagen, ct. no. 47014, Hilden, Germany) according to the manufacturer’s instructions (QIAGEN, Hilden, Germany). Then we used Nanodrop 2000 (Thermo Fisher Scientific, Waltham, MA, USA) to quantify the yields and purity of the extracted DNA. Subsequently, 16S PCR amplification and sequencing were performed using the following method described by Pereira et al. [42] with primer modifications. The V3-V4 16s region was amplified and mixed primers 341F_1–4 (CCTACGGGNGGCWGCAG) and 785R_1–4 (GACTACHVGGGTATCTAATCC), with partial Illumina TruSeq adapter sequences added to the 5′ ends (sequences of adapter presented in Appendix A). This was done in the DNA Sequencing and Genomics Laboratory of the Institute of Biotechnology of the University of Helsinki. The PCR amplification steps and MiSeq sequencing were performed according to the protocol described by Pereira et al. [42]. The MOTHUR software package was used to process the sequenced 16S rRNA gene amplicons. In the next steps, joining of two paired-end reads and demultiplexing sequences were performed, followed by quality filtering with the removal of sequences that contained bases <200 bp. The USEARCH algorithm was used to assign the operational taxonomic units (OTUs) at ≥97% similarity with chimera filtering. Then the Ribosomal Database Project (RDP) was used for the annotation of the representative sequence and for getting information on the taxonomy of each OTU.

### 2.4. Statistical Analysis

We performed data analysis using Stata 16.0 (Stata MP/16 for Windows; Stata Corp., College Station, TX, USA). In the descriptive statistics we used t-tests, and data were expressed as Least-Square Means ± SEM. Level of significance and tendency to significance were reported at a *p* value of <0.05 and <0.1, respectively. Before analysis, variables were checked whether they were normally distributed. To study the associations between farrowing data and piglet birth data at sow level variable, a sow level dataset (outcome variables measured at sow level) was used. Separate data were also used to study piglet variables after birth until seven weeks of age. After finishing the trial, sow data were merged with piglet data to study the association among sow data and weaning and post-weaning piglet data. Initially, univariate analyses with explanatory variables were performed for model building. In the full model explanatory variables with *p* ≤ 0.2 were only included and a stepwise backward elimination procedure was performed for final models. Variation inflation factor (VIF) values were used to explore the collinearity between explanatory variables and variables with VIF > 10 were removed from the model. A multivariate mixed effects model was used with post-weaning diarrhea, where piglet diarrhea at weaning and RAC were used as fixed factors, and trial batch was used as a random factor. In the case of gut microbiota analysis, we used Calypso WEB tool (http://cgenome.net:8080/calypso-8.84/faces/uploadFiles.xhtml, accessed on: 23 June 2021) [43]. In Calypso, at the preprocessing stage, we removed the samples with fewer than 1000 sequence reads. We included the maximally top 3000 taxa expressed as means later and excluded the taxa with 0.01 relative abundance across all samples. Since the gut data were not normally distributed, they were square root transformed. Subsequently, we performed analysis on correlations, ANOVA, alpha, and beta diversity indices.

## 3. Results

### 3.1. Live Birth and Total Birth

There was no significant difference in live birth piglets among groups, but there was a tendency for higher total born piglets from RAC fed sows (Control = 13.5, RAC = 15.0; *p* = 0.094). Stillborn piglet numbers were higher in the case of RAC fed sows (Control = 0.85, RAC = 0.15; *p* < 0.01).

### 3.2. Piglet Vitality

In our study, vitality score did not differ between piglets from control and RAC treated sows. The mean vitality scores for control piglets and RAC piglets were 1.39 ± 0.03, and 1.37 ± 0.03 (*p* > 0.05) respectively.

### 3.3. Colostrum Intake

Although there was no significant difference in colostrum intake (CI) between piglets born from RAC treated sows and control sows (Table 1), some differences were established according to the vitality scores of the piglets in the different treatments for different BWB. In the case of vitality 1 piglets (low vitality, *n* = 336), there were no significant differences in CI among low, medium and high body weight piglets, in RAC and control groups. However, in vitality score 2 piglets (higher vitality, *n* = 210), piglets of low birth weight (>1.0 kg) from RAC sows had significantly higher CI than piglets of the same weight from control sows (421 g vs. 214 g; *p* < 0.05; Figure 2).

### 3.4. Piglet Body Weight and Average Daily Growth (ADG)

Body weights of piglets at different stages and with different treatment combinations are shown in Figure 3. At birth, mean body weights of control piglets were higher (1534.03 g, *p* < 0.01) than those of RAC piglets (1409.65 g).

However, at weaning, there were no significant differences in piglet body weights among the four different creep treatments (CC = 8764.06 g, CR = 8776.63 g, RC = 8908.28 g, and RR = 9008.02 g, *p* = 0.21). At post-weaning, significant differences were recorded among the eight treatment groups (*p* = 0.006) at seven weeks of age, where RAC treated piglets exhibited superior growth rates (Figure 3). Piglet average daily growth (ADG) is shown in Table 2. At weaning, there were no significant differences in piglet ADG among the four different treatments (CC = 313, CR = 313, RC = 318, and RR = 322, *p* = 0.21). However, after weaning significant differences were recorded among the eight treatment groups (RRR = 306, RRC = 289, RCR = 283, RCC = 258, CRR = 278, CRC = 266, CCR = 273, and CCC = 272, *p* = 0.006;), with RAC treated piglets showing the best growth.

### 3.5. Piglet Diarrhea Incidence and Mortality Rate

In the case of piglet diarrhea at two weeks and four weeks of age, there were no significant differences between piglets that received control (CC, and RC) or RAC (CR, and RR) creep feed during lactation (*p* > 0.1). However, at seven weeks, RAC fed piglets (RRR, RCR, CRR, CCR) had significantly less diarrhea than their fellow control piglets (RRC, RCC, CRC, CCC) without RAC (*p* < 0.05) (Figure 4). In terms of mortality, there were no significant differences recorded between piglets fed with Control (CC, and RC), and RAC (CR, RR) diets until four weeks of age, and between piglets fed control (RRC, RCC, CRC, and CCC), and RAC (RRR, RCR, CRR, and CCR) diets post-weaning until seven weeks of age (*p* > 0.1; Figure 4). However, a numerically lower mortality was recorded in RAC treated piglets at seven weeks of age.

In a post-weaning piglet diarrhea model, we found that the higher the incidence of piglet diarrhea at weaning, the higher the risk for piglets to develop post-weaning diarrhea. In addition, piglets that were not fed RAC at post weaning treatment (RRC, RCC, CRC, CCC) had the higher risk of developing PWD (Table 3), whereas piglets with higher body weight at weaning had a lower risk of dying before seven weeks of age (Table 3).

### 3.6. Fecal Myeloperoxidase

MPO level tended to be higher in RAC piglets at weaning than in control piglets (C = 329.37 ng/mL, R = 419.6 ng/mL, *p* > 0.05). At post-weaning (three weeks after weaning), the mean value of MPO in control piglets was 922.59 ng/mL, which is about 175 ng/mL higher than RAC piglets (*p* < 0.05) (Figure 5).

### 3.7. Microbial Composition and Effects on Performance Parameters

Analysis of post-weaning piglet fecal samples, after quality filtering, described previously, revealed 2,998,181 reads. These reads were then analyzed for OTU assignments (at ≥97% identity level).

We measured the alpha diversity of the microbial community using Chao1, Observed and Shannon’s Indices, and no differences were established among groups (Figure 6). A total of 15 phyla were identified from the taxonomic assignments of OTUs. Firmicutes, and Bacteroides were the two dominant phyla in the fecal samples of CCC, RCC, RCR, and RRR groups. Supplementation of RAC to the post-weaning piglets decreased the abundance of Proteobacteria, Spirochaetes, Campilobacterota, Verrucomicrobia in their fecal samples (*p* < 0.05) (Figure 7). At class level, Erysipelotrichia tended to be reduced and Betaproteobacteria, Bacteroidetes__unclassified, Deltaproteobacteria, Spirochaetia and Campylobacteria were reduced by the RAC supplementation (*p* < 0.05) (Figure 7).

On the one hand, at order level, RAC supplementation reduced Bacteroidetes__unclassified, Desulfovibrionales, Spirochaetales and Campylobacterales (*p* < 0.05) (Figure 7), and tended to increase Erysipelotrichales, Acidaminococcales and Selenomonadales. On the other hand, at family level, Lachnospiraceae and Ruminococcaceae were increased, and Bacteroidetes__unclassified, Streptococcaceae, Desulfovibrionaceae, and Bacteroidales__unclassified were decreased by RAC supplementation (*p* < 0.05) (Figure 8). We identified 210 genera from the fecal samples of post weaning piglets. *Lactobacillus* was dominant followed by *Clostridium__sensu__stricto*, Lachnospiraceae__unclassified genera, *Blautia, Limosilactobacillus, Prevotella, Megasphaera, Faecalibacterium, Butyricicoccus,* Clostridiales__unclassified, *Gemmiger*, Bacteroidales__unclassified, *Mediterraneibacter, Dorea, Romboutsia*, Firmicutes__unclassified and *Holdemanella* (Table 4). We selected the 20 most abundant genera for differential analysis among four dietary groups, seven genera differed among groups (Figure 8, Appendix A). Abundance of Lachnospiraceae__unclassified was significantly higher in RAC treated piglets than control piglets. A similar result was evident for *Blautia, Butyricicoccus, Gemmiger, Mediterraneibacter*, and *Holdemanella* (*p* < 0.05) (Figure 8), whereas, Bacteroidales__unclassified was higher in CCC piglets than RRR groups (*p* < 0.05) (Figure 8).

### 3.8. Association of Microbial Genera with Piglet ADG, and Fecal MPO

From the most abundant 20 genera Blautia, Butyricicoccus, Faecalibacterium, Firmicutes__unclassified, Gemmiger, Holdemanella, Lachnospiraceae__unclassified, Prevotella, Lactobacillus, Limosilactobacillus, Mediterraneibacter, Megasphaera, and Romboutsia were negatively associated with the fecal myeloperoxidase (MPO). However, Bacteroidales__unclassified, Clostridiales__unclassified, Clostridium__sensu__stricto, Firmicutes__unclassified, Romboutsia, Ruminococcaceae__unclassified, and Terrisporobacter were positively associated with the fecal myeloperoxidase (MPO) (Table 4). In the case of ADG, Romboutsia.

Bacteroidales__unclassified, Clostridiales__unclassified, *Lactobacillus, Limosilactobacillus Clostridium__sensu__stricto*, *Terrisporobacter*, Firmicutes__unclassified, *Holdemanella* and *Mediterraneibacter* were negatively associated with ADG in post weaning piglets, whereas, Lachnospiraceae__unclassified, *Blautia, Butyricicoccus, Dorea, Faecalibacterium, Gemmiger, Megasphaera, Prevotella* and *Ruminococcus* were positively associated (Table 4).

## 4. Discussion

We determined that supplementing sow diets with RAC during late gestation and lactation, and those of their piglets in the creep and post-weaning feed, improved piglet body weight and average daily growth to the greatest extent at seven weeks of age. In addition, RAC supplementation reduced post-weaning diarrhea and fecal myeloperoxidase at that age. Low birth weight piglets with higher vitality score, born from sows which received RAC during pregnancy, were able to take in more colostrum than similar birth weight and vitality piglets from control sows. Moreover, RAC supplementation for piglets for three weeks after weaning modulated gut microbiota at phylum, order, family, and genus level.

At weaning, piglets’ intestines undergo some physiological changes followed by reduced villus height and increased crypt depth, which are associated with malabsorption [44]. This results in increased intestinal fermentation and PWD [45]. PWD has serious effects on weaned piglets, including reduced growth and increased mortality [46]. Our hypothesis was that RAC supplementation in piglet diets might modulate gut microbiota and stabilize intestinal integrity by changing the bacterial membrane macromolecules of pathogens, which might reduce the gut inflammation and MPO activity, resulting in decreased incidence of PWD. In our study, RAC supplementation for weaning piglets significantly lowered diarrhea incidence compared with the piglets fed with control feed. This finding support results from previous studies, which reported that RAC reduced the intestinal T-cells and decreased the matrix metalloproteinase (MMP), which resulted in decreasing the degrading activity of collagen type-I and collagen type-IV activity [47]. The latter are important for maintaining optimum intestinal integrity, resulting in decreased diarrhea incidence. In addition, resin acid of RAC is active mainly against the bacterial cell wall rather than cells of the host [20]. At normal pH, resin acid of RAC increases the membrane solubility and disengagement of oxidative phosphorylation, which leads to the cessation of metabolism and energy supply to the bacteria [48]. Moreover, resin acid of RAC might disrupt the bacterial cell wall and membrane structure, not only by acting as a protonophore but also by affecting cell wall permeability and fluidity by altering the cell wall macromolecules [19]. These antimicrobial actions of the resin acid might limit the abundance of pathogenic bacteria involved in PWD.

For decades, to reduce PWD incidence and improve growth performance, antibiotic growth promoters (AGP) have been useful in post-weaning piglets [49]. Due to increasing antibiotic resistance of bacteria, the EU banned the use of AGP in animal feed [8]. This raised the question of vulnerability of animals to disease, increasing the incidence of PWD as well as reducing the production efficiency and growth performance [50]. Very few alternative compounds, for example, zinc oxide, essential oils, probiotics and organic acids, are useful alternatives for controlling PWD and improving the growth performance of post-weaning piglets [51]. However, in our study, piglets that received RAC indirectly through their mother, as well as directly through creep and post-weaning feed had significantly higher ADG levels, at seven weeks of age, than piglets that had not previously been exposed to RAC. In addition, the microbiota of sows that received RAC during late gestation underwent a modulation [15] that might allow modification of the gut environment of pre- and post-weaning piglets in a manner that would improve piglet growth [52]. Moreover, RAC inhibits the growth of pathogenic bacteria that may indirectly suppress microbial competition for energy and nutrients [16] such that they might serve for piglet growth. Another RAC component, CLA, has shown anti-inflammatory and immunomodulatory effects [53]. Piglets reared by sows fed CLA during pregnancy and lactation have greater weight gain and greater final weight after weaning than pigs reared by sows fed a control diet, irrespective of the starter diet [54]. We established that piglets reared by sows fed RAC during gestation and lactation, and that received RAC in creep and post-weaning feed, had superior growth compared with control piglets that received RAC either through indirect maternal nutrition or through dietary supplementation until seven weeks of age.

Piglets fed on an RAC supplemented diet were able to develop more continuously their body weight compared with control piglets, although control piglets had higher body weight than the RAC piglets at birth. Piglets that experienced an indirect RAC effect from their mother, and received RAC directly through creep and post-weaning feed, had significantly higher body weight than control piglets at the age of seven weeks. A previous study showed that RAC supplementation to sows in late gestation enhanced colostrum yield, colostrum IgG concentration, and colostrum serum amyloid A (SAA) [15]. As a direct result, piglets were able to take in more colostrum and grew faster. This was also apparent in our study for low birth piglets with higher vitality. In the present study we found that SAA in colostrum tended to be higher in RAC treated sows. Larson et al. [55] reported that SAA stimulates the intestinal epithelium to produce mucus, which prevents the attachment of pathogenic bacteria. Because the piglets of RAC treated sows received more SAA, it might have contributed to their increase in body weight, as found in a previous study [15]. However, absorbed colostrum IgG undergoes transudation in the gut epithelium and provides immune protection against enteric infections [56], develops active immunity and improves preweaning survivability of piglets [15,57].

After weaning, piglets underwent psychological, social, environmental and dietary stresses which caused gut disturbances [58]. Gut disturbances alter enzymatic activities and intestinal architectures, resulting in intestinal inflammation [58,59]. Fecal myeloperoxidase is considered to be a potential marker for assessing intestinal inflammation [60]. In the mucosa and submucosa, neutrophilic infiltration is proportionate to the MPO activity [61]. Bontempo et al. [62] reported that *E. coli* infection elevated the MPO level in the piglet small intestine. In fattening pigs, exposure to ammonia [63] and heat stress increased fecal MPO levels by inducing inflammatory reactions [64]. However, in weaned piglets, weaning stress caused by changes in the environment and feed, as well as the immaturity of the digestive and immune systems, resulted in low feed intake and inflammation [2]. In our study, RAC supplementation significantly lowered the post-weaning piglets’ fecal MPO levels, supporting the finding that RAC inhibits pathogenic bacterial growth in chickens [39], and reduces the risk of colonization by pathogens in the pig [15]. The lowered MPO level in weaned piglets may also be due to resin acid reducing matrix metalloproteinase (MMP) activity in the small intestine, and reducing T-cell abundance in the duodenum [39,47]. In addition, RAC supplementation in weaned piglets might cause inhibition of the synthesis of proinflammatory cytokines (i.e., IL-1, IL-6, and TNF- α) at both the protein and mRNA levels via activation of the PPARγ pathway [65,66]. We found that piglets at preweaning have lower fecal MPO than at post-weaning. Our findings support those of Lemoine et al. [67] that neither weaning weight nor sex have an effect on the fecal MPO of piglets—only age. This may be because before weaning piglets can experience less exposure to a diverse microbial community due to their feeding habit, and this may lead to lower neutrophilic infiltration, resulting in low intestinal MPO. In our study, we used a commercial Pig MPO ELISA kit for detecting MPO, and our results were expressed in ng/mL according to the manufacturer’s instructions, whereas other studies used a spectrophotometer to determine MPO levels and presented the results in U/mg [64,68].

Gut microbiota play a crucial role in the gut homeostasis of piglets [32]. Our previous research showed that diet supplementation with resin acid stabilizes gut microbiota of sows [15]. In the present study we investigated the effects of RAC supplementation on piglets with RAC effects from their mothers, and that received RAC through creep feed and post-weaning feed compared with control piglets that did not receive RAC.

We recorded a low relative abundance of Proteobacteria, Spirochaetes, Campilobacterota and Verrucomicrobia in the RAC supplemented piglets. Proteobacteria are reported to be a marker for microbial dysbiosis, gut inflammation and disease [52,69]. Spirochaetes were reported to be present in the feces of diseased animals, especially in the case of swine dysentery (SD) [70,71]. The genus *Verrucomicrobia* is known for its mucus-degrading activity [72]. Some genera of the Campilobacterota are associated with gastroenteritis [73]. Effects of gut microbiota modulation at phylum level by RAC supplementation might make RAC piglets less susceptible to intestinal disease occurrences, which could be beneficial for a weaning piglet’s health. We also established that RAC supplementation increased the Lachnospiraceae and Ruminococcaceae, whereas it decreased the abundance of Streptococcaceae, Bacteroidetes__unclassified, Clostridiales__Incertae__Sedis__XIII and Desulfovibrionaceae. An increase in the Lachnospiraceae and Ruminococcaceae in the hindgut of RAC fed piglets could help them obtain more energy by using the complex polysaccharides, which are resistant to the action of digestive enzymes [74], explaining the better growth of RAC supplemented piglets in our study. The Ruminococcaceae play a particular and important role in the degradation of plant polysaccharides [32]. *Ruminococci* are considered to be important contributors in the ecosystem of the gut [32,75]. Some genera of the Desulfovibrionaceae are associated with inflammation and inhibit butyrate oxidation by their cytotoxic end product, hydrogen sulfide, resulting in impaired intestinal barrier function [76,77]. This impairment might lead to cell death and chronic inflammation [78]. Hence, RAC supplemented post-weaning piglets might have had less fecal MPO than control piglets, which is beneficial for piglet intestinal health. In addition, increased abundance of Desulfovibrionaceae might facilitate Porcine Epidemic Diarrhea Virus (PEDV) infection [77]. The *Butyricicoccus* genus of the Ruminococcaceae is known for degrading some complex carbohydrates and producing butyrate in piglets [79,80]. Butyrate is a gut-health-promoting compound that acts as the main energy source for colonocytes and exerts anti-inflammatory properties [81]. Moreover, oral administration of *B. pullicaecorum* decreased intestinal myeloperoxidase (MPO) in humans [82]. This could explain why in our study RAC piglets grew better and had less MPO. Lachnospiraceae__unclassified, *Blautia* and *Mediterraneibacter* from the Lachnospiraceae, which are were highly abundant in RAC treated piglets, are capable of degrading fiber, even cellulose and hemicellulose [83], and produce SCFA [84]. In addition, genera of the Lachnospiraceae are reported to reduce the *Clostridium difficile* in the intestine, and thereby reduce colitis incidence [85]. Bacteroidales_unclassified was less abundant in RAC piglets, and is reported to be a biomarker of diarrheic in weaning pigs [86]. Interestingly, some bacterial genera, including Lachnospiraceae__unclassified, *Blautia*, *Butyricicoccus*, and *Gemmiger*, are significantly higher in RAC supplemented piglets, and showed negative association with fecal MPO level, and positive association with ADG of post-weaning piglets. Lachnospiraceae_unclassified and *Ruminococcus* might utilize complex polysaccharides [74], and produce short chain fatty acids, especially butyrate [32,75], which promote ADG by providing energy, and maintain gut health by exerting anti-inflammatory actions [81]. Therefore, we can speculate that in response to RAC intestinal microbiota play a role in host metabolism, which hydrolyze the indigestible feedstuff, produce anti-inflammatory metabolic product, provide an additional energy source to the colonocytes and are positively correlated with ADG, and negatively with fecal MPO. Some genera, such as *Dorea*, *Faecalibacterium*, *Prevotella* and *Ruminococcus*, which were not significantly different, were negatively associated with the fecal MPO level, and positively with ADG of post-weaning piglets. These genera are reported to produce short chain fatty acids, and might promote gut health and improve ADG of post weaning piglets [87,88].

## 5. Conclusions

In conclusion, the present study supports the hypothesis that 0.1% RAC supplementation can contribute to the modulation of gut microbiota, reduce post-weaning diarrhea incidence of piglets, improve growth performance, and reduce the fecal MPO level in post-weaning piglets. If we look inside all the different treatments (namely RRR, RRC, RCR, RCC, CRR, CRC, CCR, CCC, based on the results of ADG, MPO, PWD and microbiota) it can be concluded that RRR piglet group, their mother having received RAC during late gestation (0.15%) and lactation (0.15%) and they having received RAC (0.1%) through the creep and post-weaning feed (0.1%) showed the best performances. Considering these results, RAC is a potential natural alternative to antibiotics for preventing PWD.

## Figures and Tables

**Figure 1 animals-11-02511-f001:**
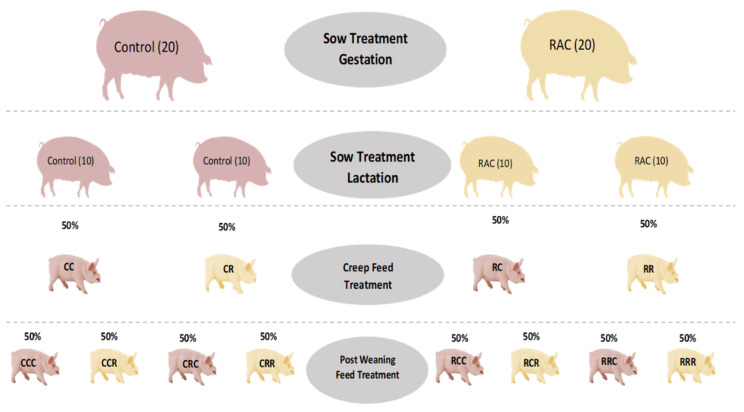
Schematic diagram of sow and piglet feeding plan. RAC = resin acid-enriched composition; RR = sow RAC feeding piglets, RAC creep feeding: RC = sow RAC feeding piglets, control creep feeding: CR = sow control feeding piglets, RAC creep feeding: and CC = sow control feeding piglets, control creep feeding. RRR = sow RAC feeding-piglets RAC creep feeding-piglets post-weaning RAC feeding: RRC = sow RAC feeding-piglets RAC creep feeding-piglets post-weaning control feeding: RCR = sow RAC feeding-piglets control creep feeding-piglets post-weaning RAC feeding: RCC = sow RAC feeding-piglets control creep feeding-piglets post-weaning control feeding: CRR = sow control feeding-piglets RAC creep feeding-piglets post-weaning RAC feeding: CRC = sow control feeding-piglets RAC creep feeding-piglets post-weaning control feeding: CCR = sow control feeding-piglets control creep feeding-piglets post-weaning RAC feeding: and CCC = sow control feeding-piglets control creep feeding-piglets post-weaning control feeding.

**Figure 2 animals-11-02511-f002:**
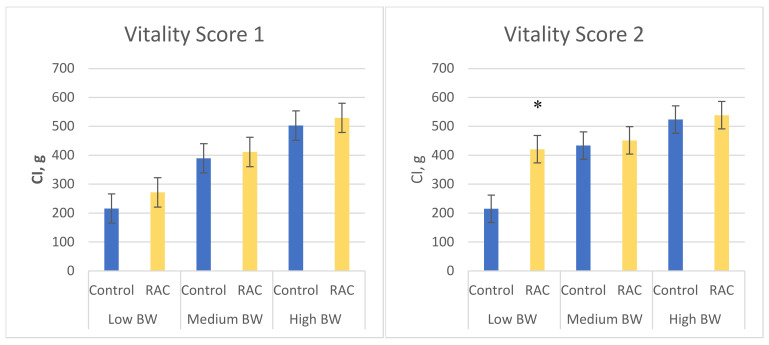
Piglet colostrum intake by vitality score, and sow treatments. Values are means ± SEMs, *n* = 236, vitality 1 piglets—no or weak movements, *n* = 210, vitality 2 piglets—body or limb movement obvious/rigorous. CI, g = colostrum intake in grams; BWB = body weights at birth; RAC = resin acid-enriched composition. * *p* < 0.05.

**Figure 3 animals-11-02511-f003:**
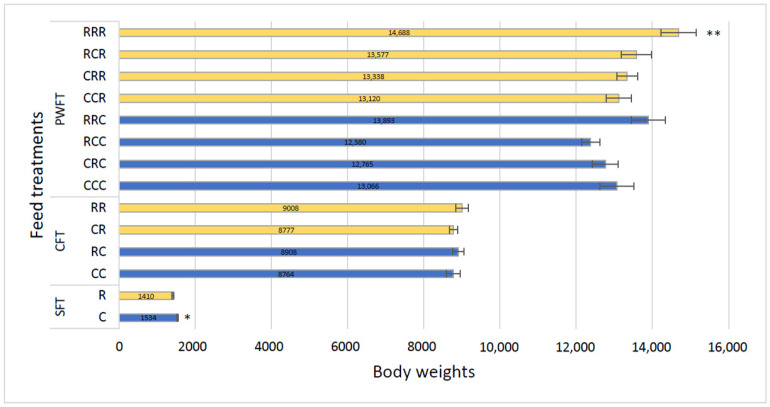
Body weights of piglets at different ages. SFT = Sow feed treatment piglets, CFT = Creep feed treatment piglets, PWFT = Post-weaning feed treatment piglets. C = Control feed of sows, R = RAC feed of sows, RR = sow RAC feeding-piglets RAC creep feeding, RC = sow RAC feeding-piglets control creep feeding, CR = sow control feeding-piglets RAC creep feeding, and CC = sow control feeding-piglets control creep feeding. RRR = sow RAC feeding-piglets RAC creep feeding-piglets post-weaning RAC feeding: RRC = sow RAC feeding-piglets RAC creep feeding-piglets post-weaning control feeding: RCR = sow RAC feeding-piglets control creep feeding-piglets post-weaning RAC feeding: RCC = sow RAC feeding-piglets control creep feeding-piglets post-weaning control feeding: CRR = sow control feeding-piglets RAC creep feeding-piglets post-weaning RAC feeding: CRC = sow control feeding-piglets RAC creep feeding-piglets post-weaning control feeding: CCR = sow control feeding-piglets control creep feeding-piglets post-weaning RAC feeding: and CCC = sow control feeding-piglets control creep feeding-piglets post-weaning control feeding.* *p* < 0.05, ** *p* < 0.01.

**Figure 4 animals-11-02511-f004:**
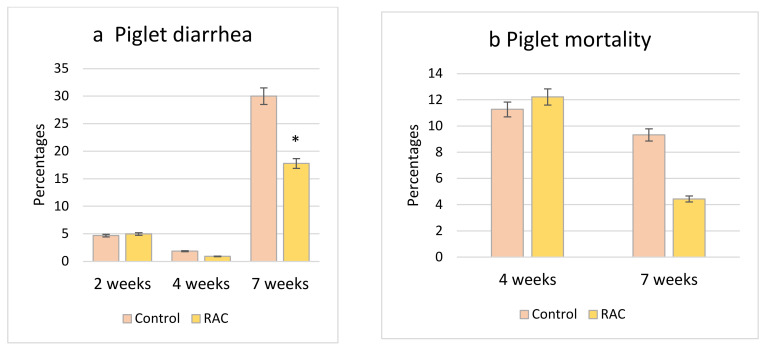
(**a**) Piglet diarrhea at two weeks, four weeks and seven weeks of age; (**b**) Piglet mortality rate at four weeks and seven weeks of age. * *p* < 0.05.

**Figure 5 animals-11-02511-f005:**
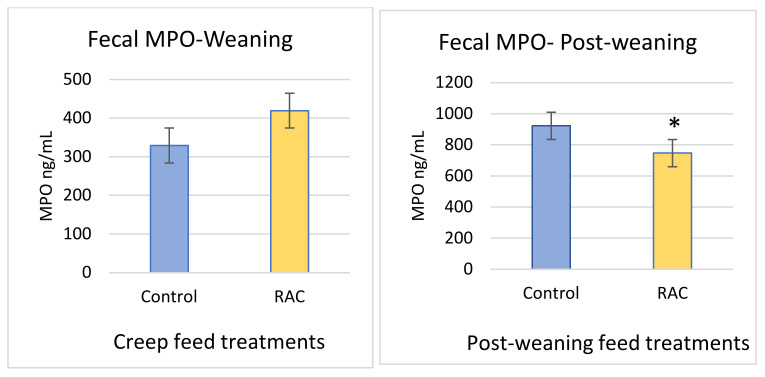
Fecal myeloperoxidase (MPO) level at weaning, and post weaning. * *p* < 0.05.

**Figure 6 animals-11-02511-f006:**
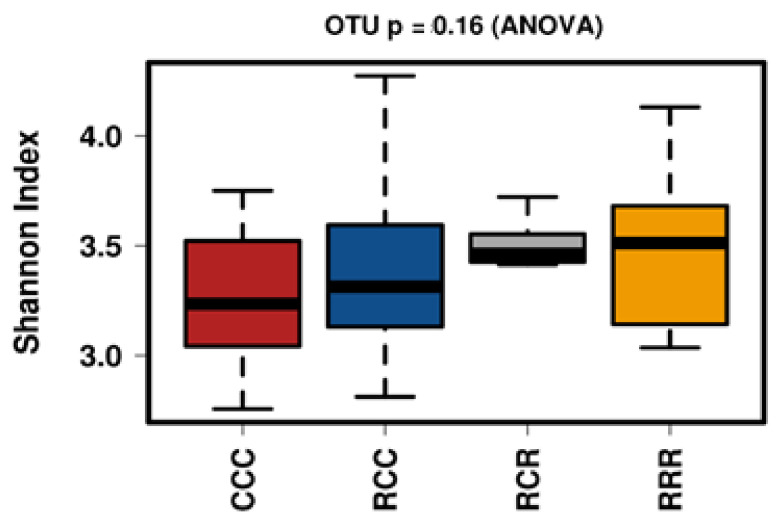
Alpha diversity-Shannon index of different feed treatment. RRR = sow RAC feeding-piglets RAC creep feeding-piglets post-weaning RAC feeding: RCR = sow RAC feeding-piglets control creep feeding-piglets post-weaning RAC feeding: RCC = sow RAC feeding-piglets control creep feeding-piglets post-weaning control feeding: and CCC = sow control feeding-piglets control creep feeding-piglets post-weaning control feeding.

**Figure 7 animals-11-02511-f007:**
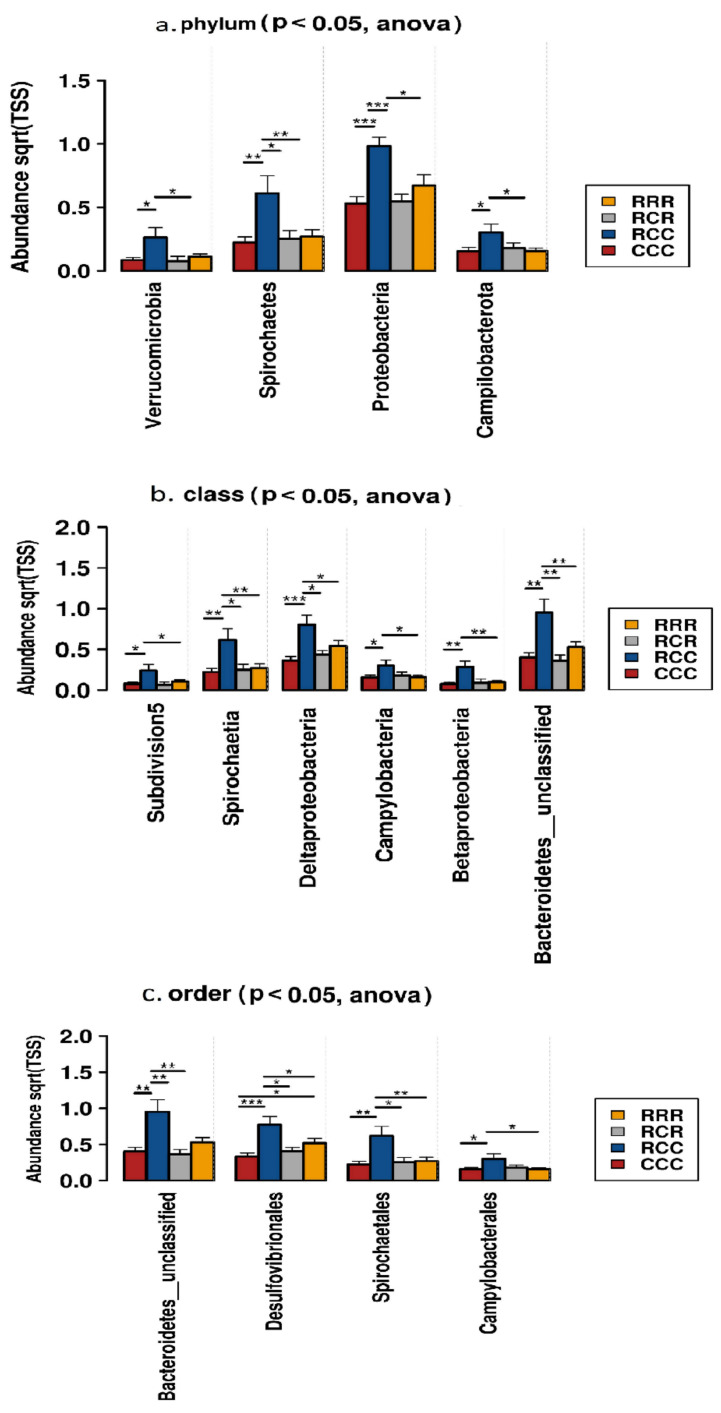
The distribution of significant bacterial taxa at (**a**) phylum; (**b**) class; and (**c**) order level in fecal samples of post weaning piglets fed different diets. RRR = sow RAC feeding-piglets RAC creep feeding-piglets post-weaning RAC feeding: RCR = sow RAC feeding-piglets control creep feeding-piglets post-weaning RAC feeding: RCC = sow RAC feeding-piglets control creep feeding-piglets post-weaning control feeding: and CCC = sow control feeding-piglets control creep feeding-piglets post-weaning control feeding. * *p* < 0.05, ** *p* < 0.01, *** *p* < 0.001.

**Figure 8 animals-11-02511-f008:**
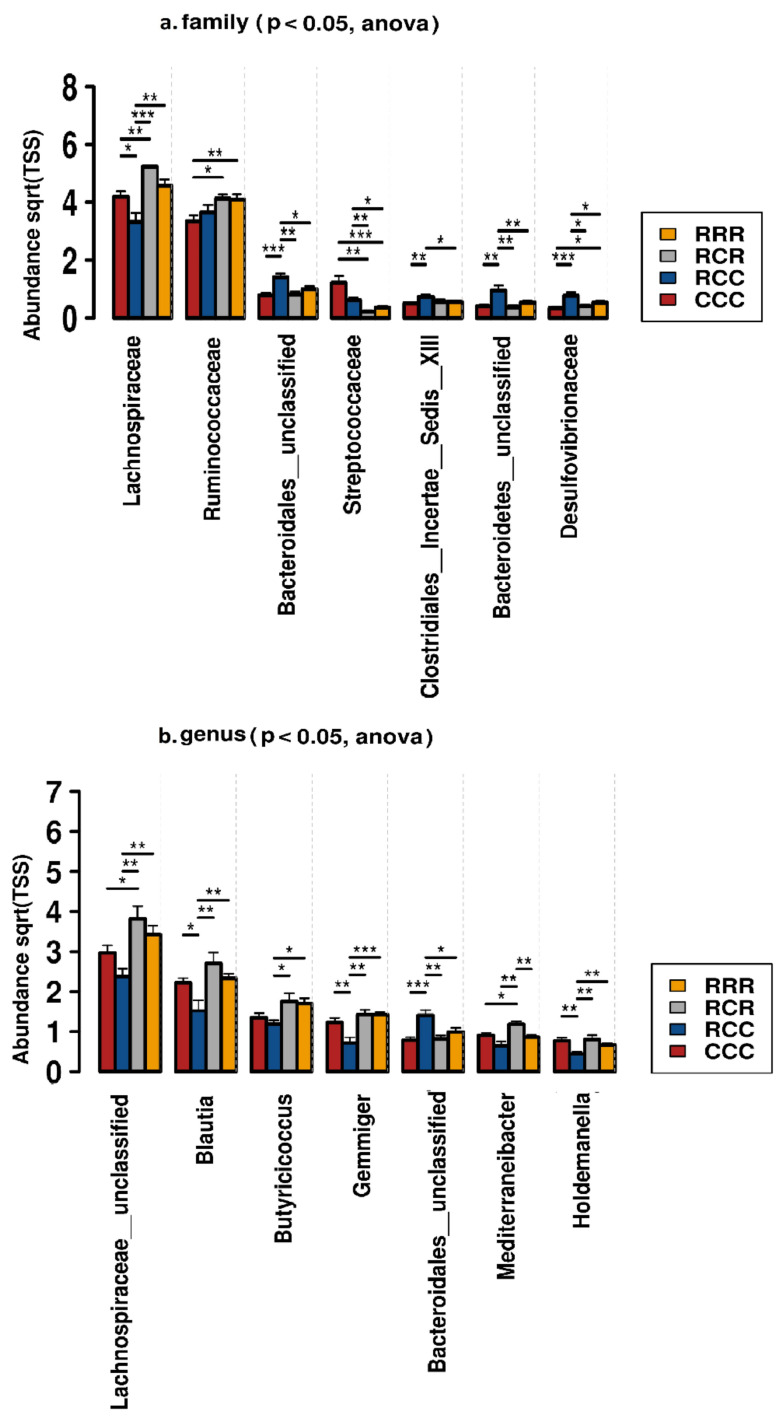
The distribution of significant bacterial taxa at (**a**) family, and (**b**) genus level in fecal samples of post weaning piglets fed different diets. RRR = sow RAC feeding-piglets RAC creep feeding-piglets post-weaning RAC feeding: RCR = sow RAC feeding-piglets control creep feeding-piglets post-weaning RAC feeding: RCC = sow RAC feeding- piglets control creep feeding-piglets post-weaning control feeding: and CCC = sow control feeding-piglets control creep feeding-piglets post-weaning control feeding. * *p* < 0.05, ** *p* < 0.01, *** *p* < 0.001.

**Table 1 animals-11-02511-t001:** Effect of dietary supplementation of sows with resin acid-enriched composition on farrowing and piglet characteristics.

Characteristics	RAC Treatment	Control	*p*-Value
Farrowing and sow characteristics			
Number of sows	20	20	
Gestation length, day	117.01 ± 0.30	117.51 ± 0.30	0.25
Farrowing duration, min	331.45 ± 41.95	296.9 ± 36.91	0.54
Sow back fat, pregnancy	10.4 ± 0.31	10.4 ± 0.33	1.00
Sow back fat, farrowing	11.65 ± 0.31	11.3 ± 0.28	0.40
Sow back fat, weaning	8.15 ± 0.16	8.2 ± 0.22	0.85
Colostrum characteristics			
Colostrum yield, g	6197.36 ± 205.03	6017.64 ± 196.50	0.53
Colostrum brix value	29.79 ± 0.78	28.36 ± 0.75	0.19
Colostrum intake, g	444.91 ± 7.86	452.44 ± 9.49	0.54
Colostrum serum amyloid A (SAA), mg/L	673.83 ± 71.47	536.73 ± 57.90	0.14
Litter characteristics			
Total born/litter size	15 ± 0.54	13.5 ± 0.67	0.09
Live-born piglets	14.15 ± 0.50	13.09 ± 0.20	0.35
Stillborn piglets	0.85 ± 0.23	0.15 ± 0.08	0.01 *
Piglet vitality	1.37 ± 0.03	1.39 ± 0.03	0.55
Weaned piglets	12.75	12.25	0.54

Data expressed as least squares means ± SEM. * Significant values at *p* < 0.05.

**Table 2 animals-11-02511-t002:** Effect of dietary supplementation of resin acid-enriched composition on piglet growth performance (ADG).

Treatment Group	Treatment	ADG	Estimate	(95% Conf. Interval)	*p*-Value
CFT (383 piglets)	CC	313			
	CR	313	0.45	−14.70–15.60	0.95
	RC	318	5.15	−10.20–20.50	0.51
	RR	322	8.71	−6.52–23.94	0.26
PWFT (285 piglets)	CCC	272			
	CCR	273	1.12	−20.71–22.95	0.92
	CRC	266	−6.27	−28.51–15.98	0.58
	CRR	278	5.65	−17.09–28.40	0.63
	RCC	258	−14.30	−35.26–6.66	0.18
	RCR	283	10.63	−11.77–33.03	0.35
	RRC	289	17.23	−5.18–39.63	0.13
	RRR	306	33.78	11.21–56.35	0.003 **

CFT = Creep feed treatment piglets, PWFT = Post-weaning feed treatment piglets. ADG = average daily growth. ** *p* < 0.01.

**Table 3 animals-11-02511-t003:** Parameter estimates of the logit model of piglet diarrhea and mortality at 7 weeks.

Outcome Variable	Predictor Variable	Odds Ratio	Confidence Interval	*p*-Value
Post-weaning diarrhea (7 weeks)	Piglet diarrhea at weaning	11.60	1.25–107.28	0.031 *
	RAC	0.52	0.29–0.93	0.027 *
Mortality at post-weaning (7 weeks)	BW at weaning			
	Small	1		0.031 *
	Average	0.20	0.02–2.46	
	Large	0.09	0.01–1.09	
	Piglet diarrhea at weaning	8.63	1.34–59.04	0.026 *

* Significant values at *p* < 0.05.

**Table 4 animals-11-02511-t004:** Effects of RAC supplementation on top 20 genera, and correlation of genus with fecal MPO and ADG of post-weaning piglets.

Taxa (Genus)	Fecal MPO		ADG		CCC Mean	RCC Mean	RCR Mean	RRR Mean	*p*-Value
	Pearson’s Index	*p*-Value	Pearson’s Index	*p*-Value					
Bacteroidales__unclassified	0.24	0.10	−0.18	0.23	0.79	1.40	0.81	0.99	0.00 *
*Blautia*	−0.12	0.42	0.33	0.02	2.22	1.52	2.70	2.33	0.00 *
*Butyricicoccus*	−0.20	0.18	0.16	0.29	1.34	1.18	1.75	1.70	0.02 *
Clostridiales__unclassified	0.10	0.49	−0.02	0.88	1.27	1.43	1.39	1.41	0.86
*Clostridium__sensu__stricto*	0.51	0.00 *	−0.20	0.18	4.22	4.23	3.00	3.30	0.14
*Dorea*	−0.27	0.07	0.10	0.51	0.86	0.69	0.86	0.80	0.30
*Faecalibacterium*	−0.30	0.04 *	0.47	0.00	1.49	1.18	2.05	1.68	0.05 *
Firmicutes__unclassified	0.30	0.04 *	−0.15	0.30	0.54	0.91	0.63	0.71	0.19
*Gemmiger*	−0.35	0.02 *	0.52	0.00	1.23	0.72	1.42	1.42	0.00 *
*Holdemanella*	−0.10	0.53	−0.09	0.57	0.76	0.43	0.80	0.66	0.01 *
Lachnospiraceae__unclassified	−0.24	0.11	0.51	0.00	2.96	2.37	3.82	3.42	0.01 *
*Lactobacillus*	−0.29	0.054	−0.03	0.84	4.64	4.40	4.10	3.95	0.51
*Limosilactobacillus*	−0.11	0.45	−0.02	0.88	2.29	2.19	1.76	2.14	0.58
*Mediterraneibacter*	−0.15	0.33	−0.04	0.78	0.90	0.64	1.18	0.86	0.00 *
*Megasphaera*	−0.35	0.02 *	0.20	0.19	1.42	1.59	1.75	1.78	0.70
*Prevotella*	−0.36	0.01 *	0.27	0.07	1.51	1.67	2.13	2.25	0.27
*Romboutsia*	0.43	0.00 *	−0.33	0.02	0.72	1.02	0.64	0.57	0.13
Ruminococcaceae__unclassified	0.23	0.13	−0.04	0.78	1.88	2.64	2.10	2.48	0.05 *
*Ruminococcus*	−0.20	0.18	0.14	0.36	0.84	0.98	1.13	1.03	0.33

* Significant values at *p* < 0.05.

## Data Availability

The data that support the findings presented in this experiment are available upon direct request to the corresponding author.

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
