# Peer review of "In-Feed Supplementation of Resin Acid-Enriched Composition Modulates Gut Microbiota, Improves Growth Performance, and Reduces Post-Weaning Diarrhea and Gut Inflammation in Piglets"

_animals, 2021, doi:10.3390/ani11092511_

Round 1

Reviewer 1 Report

Reviewer's remarks / comments

Comment 1 concerns the chapter
2. Materials and Methods
2.1. Animals and experimental design

There is no information on the characteristics of the RC product and its source of origin.
How RC was supplemented in the diets of sows and piglets ?
The information given on line 132-133 ,, The creep feed diet formulation is described in the S2 Text ‘’ requires clarification, especially the term S2 Text- difficult to find in the text ?

The same is true on line 139-140.
Comment 2 concerns the chapter

5. Conclusions
Specific doses of RC were used in the research, in the summary of the article, I think that they should be indicated and should one refer to the size of the RC doses used?

Comment 3 concerns the chapter
References
Shorten the list of references numerically, take into account from the last 10-12 years.

Author Response

Dear Reviewer,

Thank you for making useful comments that have certainly improved the original manuscript. Following you will find the detailed response to your comments.

-Comment 1 concerns the chapter
2. Materials and Methods
2.1. Animals and experimental design

-There is no information on the characteristics of the RC product and its source of origin.
-How RC was supplemented in the diets of sows and piglets?
-requires clarification, especially the term S2 Text- difficult to find in the text?

The information given on lines 132-133, The creep feed diet formulation is described in the S2 Text ‘’. Detailed of the RAC was already described in the L66-70. However, for more clarification, we have mentioned the commercial name and producer of the product in the L122-123 and added a supplementary table (S3 text) with the detailed composition of the product.

We mentioned RAC supplementation in gestation and lactation diets in L122-123 and L125-126 respectively, and for piglets in creep feed and post-weaning feeding section. This means RAC was already a part of the compound feed formulation, more specially spraying during the feed mixing. 

We have removed the sentence L132-33. To make it more specific we have added the S2 Text now in L130-31.

-The same is true on line 139-140.
Comment 2 concerns the chapter

We have removed the sentence L139-40. To make it more specific we have added the S2 Text now in L138).

-5. Conclusions
Specific doses of RC were used in the research, in the summary of the article, I think that they should be indicated, and should one refer to the size of the RC doses used?

We have updated the information now in the conclusion in L510 for creep feed and in L515 for sow and post-weaning feed doses of RAC.

-Comment 3 concerns the chapter References Shorten the list of references numerically, consider from the last 10-12 years.

Answer -We have shortened the references list now and removed 24 of the really old references wherever possible.

Reviewer 2 Report

Dear Author, you have an interesting and quite complete manuscript. However, there are a few points that can be improved:

  1. There are a few typos around the text
  2. The figures quality can be improved; graphics seems to be quite small
  3. Do you have any pictures showing the histology of the intestine? IT should be added, since this is important to understand the phenotype in each treatment, and also this point is discussed in the text
  4. On line 380, 381, 383 you refer to RA. This anacronyms has not been cited before in the manuscript. Is it a RAC anacronyms mispealed?
  5. On the conclusion, I believe that it should be interesting if the author can conclude, comparing the 8 treatments, about the best treatment to be used to improve animal production 

Author Response

Dear Reviewer,

Thank you for making useful comments that have certainly improved the original manuscript. Following you will find the detailed response to your comments.

-There are a few typos around the text

We have checked the text thoroughly and corrected the typos.

-The figures quality can be improved; graphics seems to be quite small

We have updated the figures now and made the graphics relatively bigger.

-Do you have any pictures showing the histology of the intestine? IT should be added, since this is important to understand the phenotype in each treatment, and also this point is discussed in the text

Our study didn’t have an ethical permission for euthanizing an animal for collecting organ. However, histology was beyond the scope of this study.

-On line 380, 381, 383 you refer to RA. This anacronyms has not been cited before in the manuscript. Is it a RAC anacronyms mispealed?

We have corrected in the in L381,382, and 384. and acronyms are cited in correct ways.

-On the conclusion, I believe that it should be interesting if the author can conclude, comparing the 8 treatments, about the best treatment to be used to improve animal production 

We have updated the conclusion now and compared the treatments and mentioned the best treatment with adding RAC in the diet according to our results in L513-17.